# Mn^2+^ Luminescence in Ca_9_Zn_1–*x*_Mn*_x_*Na(PO_4_)_7_ Solid Solution, 0 ≤ *x* ≤ 1

**DOI:** 10.3390/ma16124392

**Published:** 2023-06-14

**Authors:** Eldar M. Gallyamov, Vladimir V. Titkov, Vladimir N. Lebedev, Sergey Y. Stefanovich, Bogdan I. Lazoryak, Dina V. Deyneko

**Affiliations:** 1Department of Chemistry, Lomonosov Moscow State University, 119991 Moscow, Russia; e.m.gallyamov@gmail.com (E.M.G.); vlatitkov@yandex.ru (V.V.T.); vladimir.lebedev@chemistry.msu.ru (V.N.L.); bilazoryak@gmail.com (B.I.L.); deynekomsu@gmail.com (D.V.D.); 2Laboratory of Arctic Mineralogy and Material Sciences, Kola Science Centre, Russian Academy of Sciences, 184209 Apatity, Russia

**Keywords:** phosphate, β-Ca_3_(PO_4_)_2_, TCP, red luminescence, manganese, crystal structure

## Abstract

The solid solution Ca_9_Zn_1–*x*_Mn*_x_*Na(PO_4_)_7_ (0 ≤ *x* ≤ 1.0) was obtained by solid-phase reactions under the control of a reducing atmosphere. It was demonstrated that Mn^2+^-doped phosphors can be obtained using activated carbon in a closed chamber, which is a simple and robust method. The crystal structure of Ca_9_Zn_1–*x*_Mn*_x_*Na(PO_4_)_7_ corresponds to the non-centrosymmetric β-Ca_3_(PO_4_)_2_ type (space group *R*3*c*), as confirmed by powder X-ray diffraction (PXRD) and optical second-harmonic generation methods. The luminescence spectra in visible area consist of a broad red emission peak centered at 650 nm under 406 nm of excitation. This band is attributed to the ^4^T_1_ → ^6^A_1_ electron transition of Mn^2+^ ions in the β-Ca_3_(PO_4_)_2_-type host. The absence of transitions corresponding to Mn^4+^ ions confirms the success of the reduction synthesis. The intensity of the Mn^2+^ emission band in Ca_9_Zn_1–*x*_Mn*_x_*Na(PO_4_)_7_ rising linearly with increasing of *x* at 0.05 ≤ *x* ≤ 0.5. However, a negative deviation of the luminescence intensity was observed at *x* = 0.7. This trend is associated with the beginning of a concentration quenching. At higher *x* values, the intensity of luminescence continues to increase but at a slower rate. PXRD analysis of the samples with *x* = 0.2 and *x* = 0.5 showed that Mn^2+^ and Zn^2+^ ions replace calcium in the *M*5 (octahedral) sites in the β-Ca_3_(PO_4_)_2_ crystal structure. According to Rietveld refinement, Mn^2+^ and Zn^2+^ ions jointly occupy the *M*5 site, which remains the only one for all manganese atoms within the range of 0.05 ≤ *x* ≤ 0.5. The deviation of the mean interatomic distance (∆*l*) was calculated and the strongest bond length asymmetry, ∆*l* = 0.393 Å, corresponds to *x* = 1.0. The large average interatomic distances between Mn^2+^ ions in the neighboring *M*5 sites are responsible for the lack of concentration quenching of luminescence below *x* = 0.5.

## 1. Introduction

Currently, there is a lot of research being devoted to phosphors, which are effective for indoor LED lighting when combined with InGaN diodes [1,2,3]. In the structural family of mineral whitlockite, phosphors with the β-Ca_3_(PO_4_)_2_-type may contain europium or manganese as guest atoms, which exhibit red emission under InGaN excitation. In some inorganic hosts, the emission bands in the red area of the visible spectra are weak, which limits their applications. Thus, phosphors with the β-Ca_3_(PO_4_)_2_ type structure have attracted growing attention, due to their favorable parameters of transparency, environmental friendliness, biocompatibility, and biodegradability. 

Phosphors doped with Mn^2+^ ions are of particular interest currently due to the broad red band in the photoluminescence (PL) spectra [4]. Manganese ions may be stabilized in the divalent state with a wide range of concentrations in the special sites in the β-Ca_3_(PO_4_)_2_-type structure. In the β-Ca_3_(PO_4_)_2_ (β-TCP) crystal structure, there are five crystallographic independent cation sites (*M*1–*M*5), each of the *M*1, *M*2, *M*3 and *M*5 sites is fully occupied by Ca atoms [5]. The *M*4 site has a variable occupation, ranging from fully vacant to fully occupied. The occupation of the *M*4 site in the initial β-TCP host is responsible for the wide range of isomorphism. 

The luminescence properties are significantly affected by the coordination of Mn^2+^ ions. Thus, when Mn^2+^ is located in the strong crystal field of the octahedral coordination, β-Ca_3_(PO_4_)_2_:Mn^2+^ phosphor shows a wide intense red band in the PL spectra at 570–750 nm centered at 650 nm [4]. An important crystal chemistry factor for stabilizing of Mn^2+^ is its location in sites with octahedral coordination in the β-TCP structure, along with other divalent cations of a similar size, such as zinc or magnesium. When manganese ions change their charge, the resulting emission color can shift to blue or green. It should be noted that manganese ions with other oxidation states other than two positive, cannot occupy the octahedral *M*5 site of the β-TCP structure. According to previous studies, Mn^3+^ ions can occupy *M*1–*M*3 sites [6], while Mn^4+^ can only be in tetrahedral coordination. However, there is no reference data on the PO_4_ → MnO_4_ substitution in the tetrahedral sites of the β-TCP structure.

Satisfying the condition of the Mn^2+^ occupation, its concentration in the β-TCP-type structure can vary significantly without structural disorder or charge compensation. To stabilize Mn^2+^ within the host, a reduction atmosphere is necessary during solid-state synthesis to prevent oxidation. The most common procedure involves using a reduction gas, such as a mixture of N_2_ + H_2_ or H_2_ only [7,8,9,10,11]. Therefore, the use of “soft” reduction methods is a promising technique. 

Our main interest is to control the occupancy of nonequivalent cationic sites [12] by Mn^2+^ ions in the β-Ca_3_(PO_4_)_2_-type structure. Kostiner and Rea [13] first reported that Ca^2+^ occupies the *M*1–*M*3 positions, while the *M*4 site is empty or partly occupied with single or divalent cations. The six-fold oxygen-coordinated *M*5 position is always occupied by di- and trivalent cations of an appropriate size, equal to or smaller than calcium. Mg^2+^ cations are found only in the *M*5 position [7,14] or are distributed over the *M*4 and *M*5 sites [15]; Zn^2+^ cations are known to settle in the *M*5 site [8]. Eu^2+^_,_ Ce^3+^, and other trivalent rare earth element (REE) ions are often used as sensitizers. In Mn^2+^-doped β-TCP phosphors, Eu^2+^ and Ce^3+^ ions are predominantly distributed in the *M*1–*M*3 sites, while Mn^2+^ is concentrated in the *M*5 site [5,6,8,12,13,14]. To exclude the *M*4 site from the number of possible positions for manganese ions, this position can be filled with a combination of mono- and divalent cations of alkali or alkaline earth metals [11,13]. When the *M*4 site is occupied by K^+^ ions [12], both Mn^2+^ and Eu^2+^ ions occupy any of the *M*1, *M*2, *M*3, or *M*5 sites. As a result, in the PLE spectra of Ca_10_K(PO_4_)_7_:Eu^2+^/Mn^2+^ and Ca_9_MnNa(PO_4_)_7_ [9,10], the excitation peaks of Mn^2+^ are related only to the *M*5 position. On the other hand, this position is exclusively preferred by Zn^2+^ in Ca_8_ZnLn(PO_4_)_7_ [16] and in the whole range of the Ca_9−*x*_Zn*_x_*Gd_0.9_(PO_4_)_7_:0.1Eu^3+^ solid solution [17]. Although all Ca_9−*x*_Zn*_x_*Gd_0.9_(PO_4_)_7_:0.1Eu^3+^ compositions belong to the β-Ca_3_(PO_4_)_2_-type structure, their structure changes from polar at *x* = 0 (space group *R*3*c*) to centrosymmetric (space group *R*$\bar{3}$*c*) with increasing *x* values. The phosphate Ca_8_ZnGd_0.9_Eu_0.1_(PO_4_)_7_ acts as the most effective phosphor in this solid solution.

The effective luminescence of manganese ions in the β-Ca_3_(PO_4_)_2_ host depends on a set of factors. The most important is the optimal arrangement of Mn^2+^ ions in the crystal structure. Optimization of the concentration and methods of sensitization is an urgent problem. The solution to this problem should lead to the most effective phosphors. At the same time, it is important to determine reliable and simple conditions to obtain the β-Ca_3_(PO_4_)_2_:Mn^2+^ phosphors with a well-ordered crystal structure and minimal defects. Several related works have shown that Zn^2+^ ions can improve the luminescence intensity of the β-TCP-type phosphors [18]. This effect has been explained by the distortion of the local environment of the luminescence centers [19]. It should be noted that similar regularities have also been observed in other types of phosphors, for example, in molibdates [20]. The positive effect of the presence of Zn^2+^ in the crystal structure concerns not only the substitutions in the host lattice, but also the anionic substitutions [21]. 

For the purpose of the orderly placement of all atoms, Zn^2+^ ions [16,17,19] were chosen as the main filling atom for the *M*5 octahedra. Na^+^ ions were used for filling the *M*4 sites in the β-TCP structure. In this work, we explored the synthesis of Ca_9_Zn_1−*x*_Mn*_x_*Na(PO_4_)_7_ phosphates as well as a detailed X-ray diffraction analysis of all oxygen polyhedra in their crystal structures, paying special attention to the (Mn/Zn)O_6_ octahedra in connection with luminescence properties.

## 2. Materials and Methods

### 2.1. Synthetic Procedure

Ca_9_Zn_1**−***x*_Mn*_x_*Na(PO_4_)_7_ with *x* = 0.05, 0.1, 0.2, 0.4, 0.5, 0.7, and 1.0 (for short CZ_1–*x*_M*_x_*N) was synthesized by a standard solid-state method from stoichiometric mixtures of CaCO_3_, ZnO, NH_4_H_2_PO_4_, MnCO_3_, and Na_2_CO_3_ reagents purchased from Sigma-Aldrich [6]. Firstly, the reagents with a chemical purity of 99.9% were heat-treated to remove possible moisture impurities. Moreover, this was performed to reliably eliminate the possible admixtures and maximize purity of the raw materials. After that, the stoichiometric mixtures were heated to 1373 K and kept for 24 h in a reducing atmosphere of activated carbon to enable the reaction. Powder X-ray diffraction patterns of the synthesized powders were checked using the ICCD PDF-4 database to ensure the absence of all extraneous reflections of the initial or intermediate phases.

### 2.2. Powder X-ray Diffraction Study 

Powder X-ray diffraction (PXRD) patterns were collected on Rigaku SmartLab SE (Rigaku Corporation, Wilmington, MA, USA) (3 kW sealed X-ray tube; D/teX Ultra 250 silicon strip detector; vertical type θ-θ geometry; HyPix-400 (2D HPAD) detector in the 2θ range between 3° and 110° with a step interval of 0.02°). The LeBail decomposition was applied using the JANA2006 software [22]. The crystal structure was successfully refined by the Rietveld method using JANA2006 software [22]. Illustrations were created with DIAMOND [23] software.

### 2.3. Second Harmonic Generation Study 

The second harmonic generation (SHG) method was applied for controlling the space group of the samples. Doubling the frequency of laser radiation allows us to reliably distinguish non-centrosymmetric phases from centrosymmetric phases with help SHG effect [24]. Previously, we successfully used this method to distinguish the phosphates with or without center of symmetry as well as to reveal transformations of non-centrosymmetric phases into centrosymmetric ones [17]. The second harmonic signal was produced in the as-synthesized powders by radiation of Q-switched YAG:Nd laser at λ = 1064 nm and measured in the reflection mode in comparison with SHG response from α-SiO_2_ powder, which was used as a reference.

### 2.4. Photoluminescence Study

The luminescence study (PLE and PL spectra) was carried out on an Agilent Cary Eclipse fluorescence spectrometer (Agilent Technologies, Santa Clara, CA, USA) with a 75 kW xenon light source (pulse duration τ = 2 μs; pulse frequency ν = 80 Hz; wavelength resolution 0.5 nm; a PMT Hamamatsu R928). All of the spectra were obtained under similar experimental conditions for a correct comparison of the luminescence intensities. The obtained PLE spectra were corrected on the sensitivity of the spectrometer.

## 3. Results

### 3.1. Structural Study

The PXRD patterns of all the synthesized CZ_1**–***x*_M*_x_*N (0 ≤ *x* ≤ 1.0) samples match the Ca_9.5_Zn(PO_4_)_7_ compound with the β-TCP-type structure [15]. This proves that all the loaded elements in the stoichiometric ratios are completely incorporated into the structure in accordance with the applied Ca_9_Zn_1**−***x*_Mn*_x_*Na(PO_4_)_7_ formulas in the entire range of 0 ≤ *x* ≤ 1.0. It can also be suggested that Mn^2+^ and Zn^2+^ occupy *M*5 sites in the β-Ca_3_(PO_4_)_2_-type structure. The regular shift in X-ray reflections in the solid solution reveals that replacing zinc with manganese generally increases all the unit cell parameters, but this shift is a nonlinear function of *x*.

To clarify the structural feature of the Zn^2+^ → Mn^2+^ substitution, the crystal structures were examined using data from precision PXRD experiments for the Ca_9_Zn_1**−***x*_Mn*_x_*Na(PO_4_)_7_ compositions with *x* = 0.05, 0.1, 0.2, 0.4, and 0.5. The PXRD patterns are shown in Figure 1. The main Crystallographic data on the structure refinement are listed in Appendix A. Appendix A list the fractional atomic coordinates, site-occupancy factors, and isotropic atomic displacement parameters. Selected interatomic distances are shown in Appendix A. The results of the Rietveld refinement of the structures are available as CCDC 2256115, 2256116, 2256117, 2256118, and 2256119 cards. 

The unit cell parameters and volumes of the Ca_9_Zn_1**–***x*_Mn*_x_*Na(PO_4_)_7_ solid solution are shown in Figure 2. The dependence of the parameters *a*, *c*, and *V* on *x* is not monotonic and demonstrates an obvious negative deviation from the Vegard’s law. Our structural data (see below) point to the crucial role of the ordering of the metal–oxygen bonds in oxygen polyhedra around large cations in the crystal lattice of the Ca_9_Zn_1**−***x*_Mn*_x_*Na(PO_4_)_7_ solid solution, which might explain this dependence. Suspicions of possible phase rearrangements were rejected in light of the smooth behavior of the SHG intensity everywhere when 0 ≤ *x* ≤ 1.0, roughly corresponding to the SHG response of the quartz reference.

The structure of Ca_9_MnNa(PO_4_)_7_ (PDF-4 00-050-0214, space group *R*3*c*) served as the starting model for the Rietveld refinement for the Ca_9_Zn_1**−***x*_Mn*_x_*Na(PO_4_)_7_ solid solutions, where *x* = 0.05, 0.1, 0.2, 0.4, and 0.5. The analysis showed that the *M*1, *M*2, and *M*3 sites are fully occupied by Ca^2+^ cations. The Na^+^ cations were placed in the *M*4 site, while the mixed Mn^2+^/Zn^2+^ cations occupied the *M*5 position. Further, the coordinates of all atoms were refined, taking into account their occupancy. It was found that the atomic coordinates and occupancy of all positions were fully consistent with the structural type of the β-Ca_3_(PO_4_)_2_.

The joint filling of all oxygen octahedra by Mn^2+^ and Zn^2+^ ions excludes their presence in other possible positions or a redistribution between them with a change in concentration. The resulting structure demonstrates an ordered arrangement of atoms (Figure 3a). However, the crystal lattice allows for the formation of structural defects due to the disturbance of the regularity of metal–oxygen bonds.

The structure of Ca_9_Zn_1**−***x*_Mn*_x_*Na(PO_4_)_7_ is composed of five oxygen polyhedra: *M*1, *M*2, *M*3, *M*4, and *M*5 (Figure 3a). They are connected by a bridging oxygen atom or common edge. Hence, any increase in the <*M*–O> distance due to *x* variations causes deformations in all the adjacent polyhedra. Table 1 shows the mean interatomic distances in the *M*1–*M*5 polyhedra and P1–P3 tetrahedra.

The experimental data in Table 1 do not reveal noticeable changes in the mean <*M*–O> bond lengths with varying *x*. Furthermore, the mean distances in the polyhedra do not explain the concentration anomalies of the unit cell parameters. Thus, we performed a detailed analysis of the oxygen octahedron around the *M*5 site (Figure 3b), taking into account the fact that this is a site for substitution by cations with close but still different ionic radii, *r*_VI_ = 0.83 (Mn^2+^) and *r*_VI_ = 0.74 (Zn^2+^) Å, respectively [26].

The lengths of nonequivalent *M*–O_6_ and *M*–O_9_ bonds in the *M*5 octahedra were calculated from the coordinates of the corresponding atoms in Ca_9_Zn_1**–***x*_Mn*_x_*Na(PO_4_)_7_ with *x* = 0.05, 0.1, 0.2, 0.4, and 0.5 and compared with those available in the literature data. Since the distances in the polyhedra for Ca_9_ZnNa(PO_4_)_7_ have not been reported previously in the reference data, a close composition of Ca_9.5_Zn(PO_4_)_7_ was taken as the zero point of *x* = 0. The difference in the Zn–O_6_ and Zn–O_9_ bonds in this composition is ∆*l* = 0.13 Å. The end point of *x* = 1 corresponds to Ca_9_MnNa(PO_4_)_7_ [11] with the strongest bond length asymmetry being ∆*l* = 0.393 Å, which is the difference between the Mn–O_6_ and Mn–O_9_ bond lengths.

The data in Table 2 demonstrate a significant change in the shape of the oxygen octahedra around the *M*5 sites depending on *x*. According to these data, the asymmetry of metal–oxygen bonds in the octahedra decreases with the introduction of Mn^2+^ down to ∆*l* = 0.003 Å at *x* = 0.2. This pattern corresponds to octahedra strongly elongated along the *c*-axis at the extreme points of the solid solution and compressed to an almost isotropic shape with average *x* values. Such behavior of the octahedra shape in the solid solution should be accompanied by a similar symmetrization of all other oxygen polyhedra and hence with a reduction in the unit cell parameters at intermediate *x* values. 

### 3.2. Photoluminescence Study

Figure 4 shows the PLE (a) and PL (b) spectra of the Ca_9_Zn_1–*x*_Mn*_x_*Na(PO_4_)_7_ solid solutions. On the PLE spectra monitored at λ_em_ = 642 nm, a series of bands are observed, related to the transitions from the ^6^A_1_(^6^S) ground state to the ^4^E(^4^D) (360 nm), ^4^T_2_(^4^D) (370 nm), ^4^A_1g_^4^E_g_ (406 nm), and ^4^T_1g_ (455 nm) excited levels of Mn^2+^ ions. These bands are attributed to the d-d electron transitions of Mn^2+^, and their location is in accordance with previous data on Mn^2+^ excitation in β-Ca_3_(PO_4_)_2_-type phosphors [5]. It should be noted that in the Ca_9_Zn_0.9_Mn_0.1_Na(PO_4_)_7_ host, the charge transfer band (CTB) is weakly pronounced (Figure 4a). The CTB is commonly observed around 250 nm in Mn^2+^-doped β-Ca_3_(PO_4_)_2_-type phosphates [7] and corresponds to the O^2−^ → Mn^2+^ transition of excited 2p^6^ electrons from the orbital of the O^2−^ ion to the incomplete 3d orbital of Mn^2+^. All samples from the Ca_9_Zn_1**−***x*_Mn*_x_*Na(PO_4_)_7_ solid solution show the nearly similar excitation spectra differing only in the intensity of the bands (Figure 4a). This fact indicates that, in all the studied samples, the Mn^2+^ ions are excited by the same excitation bands.

The PL spectra of Ca_9_Zn_1–*x*_Mn*_x_*Na(PO_4_)_7_ shows a broad band at 600–750 nm centered at 642 nm (Figure 4b). This band is related to the spin-forbidden ^4^T_1_ → ^6^A_1_ transition. The emission color of Mn^2+^ is strongly dependent on the crystal field strength and the coordinational environment [27,28,29,30]. The absence of a blue shift in the PL spectra (Figure 4b) confirms the PXRD data: in all the samples from the Ca_9_Zn_1–*x*_Mn*_x_*Na(PO_4_)_7_ solid solution, the Mn^2+^ ions are located in the *M*5 site of the structure. Note that there are no sharp lines attributed to Mn^4+^, which shows the absence of oxidation processes during synthesis in air. In samples with *x* = 0.1 and 0.2, an additional broad band is observed at 500–600 nm (Figure 4b) attributed to defect-related bands.

Figure 5a shows the concentration dependence of the emission intensity of Mn^2+^ in the Ca_9_ZnNa(PO_4_)_7_ host. A linear rise in intensity is observed at 0.05 ≤ *x* ≤ 0.5. At *x* > 0.5, the emission intensity of the ^4^T_1_ → ^6^A_1_ band is still increasing; however, it deviates from a straight line.

The CIE 1931 chromaticity coordinates (*x*, *y*) for Ca_9_Zn_1–*x*_Mn*_x_*Na(PO_4_)_7_ were calculated from the spectra according to formula presented in [31]. Figure 5b shows the CIE coordinates under λ_ex_ = 406 nm excitation. The samples with *x* = 0.1 and 0.2 show an orange color for the emission according to the superposition of broad bands at 500–575 nm and 600–700 nm (Figure 4b). With the increase in the Mn^2+^ concentration, the band centered at 650 nm becomes dominant, and the samples when *x* ≥ 0.4 show red emissions (Figure 5b). The values of the CIE coordinates are listed in Appendix A.

## 4. Discussion

The combination of the structural and optical characteristics of the Ca_9_Zn_1**−***x*_Mn*_x_*Na(PO_4_)_7_ samples obtained under mild reductive conditions using activated carbon indicates the existence of a single-phase solid solution in the 0 ≤ *x* ≤ 1.0 range. Moreover, the constituent gases in the closed chamber according to the disappearance of charcoal during the synthesis might be residual N_2_ as well as CO_2_ or CO. In particular, PXRD patterns for Ca_9_Zn_1**−**_*_x_*Mn*_x_*Na(PO_4_)_7_ with 0.05 ≤ *x* ≤ 0.5 correlated with other β-Ca_3_(PO_4_)_2_-type compounds [11]. All the diffraction peaks demonstrate a good agreement with the reference and the absence of impurity phases. Based on this, one may conclude that all manganese ions in Ca_9_Zn_1**−**_*_x_*Mn*_x_*Na(PO_4_)_7_ were completely incorporated into the β-Ca_3_(PO_4_)_2_-type structure [32].

The crystal lattice of the Ca_9_Zn_1**−**_*_x_*Mn*_x_*Na(PO_4_)_7_ solid solution slightly expands with increases in *x*, when the larger Mn^2+^ ion is substituted with the smaller one (Zn^2+^). At the same time, on the concentration dependences of the unit cell parameters a(*x*) and c(*x*), a region of 0.05 < *x* < 0.5 was found, in which these concentration dependences have a negative deviation from the Vegard’s law. The structural refinement using the Rietveld method for the samples whith 0.05 ≤ *x* ≤ 0.5 showed that Mn^2+^ ions substitute Zn^2+^ ions statistically in the octahedral oxygen-coordinated *M*5 site in the β-Ca_3_(PO_4_)_2_ crystal structure. This position remains the only one for all the manganese and zinc atoms. The joint occupation of Zn^2+^/Mn^2+^ in the *M*5 sites results in Ca_9_Zn_1**−**_*_x_*Mn*_x_*Na(PO_4_)_7_ samples with 0.05 ≤ *x* ≤ 0.5 and leads to a decrease in the anisotropy of the length of metal–oxygen [25]. The interatomic distances of the *M*1–*M*4 sites slightly change with variations in the Mn^2+^ concentration in the whole range of the solid solution existence. This structural feature indicates a positive effect of the joint statistical occupation of the octahedra by Zn^2+^ and Mn^2+^ ions on both the octahedra shape and apparently on the ordering of their nearest atomic environment. Thus, the minimum number of structural defects in this environment appears.

Upon excitation at 406 nm, the PL spectra consist of a broad band in the red region due to the ^4^T_1_ → ^6^A_1_ electron transition of Mn^2+^, which is characteristic for the d-d transition. Defect-related bands at 500–600 nm in the PL spectra are observed only for samples with a low Mn^2+^ content (*x* ≤ 0.2) in the Ca_9_Zn_1**−***x*_Mn*_x_*Na(PO_4_)_7_ compounds. This fact is explained by the maximum defect formation in the solid solution related to the size mismatch between the Mn^2+^ and Zn^2+^ ions at the small substitution level. Then this effect disappears due to the averaging of the distribution of Mn^2+^ and Zn^2+^ over the *M*5 sites.

The broad band in the PLE spectra at 450 nm and intensive band at 403 nm indicate the possibility of using both blue and UV chips as the excitation sources for Ca_9_Zn_1**−***x*_Mn*_x_*Na(PO_4_)_7_ solid solution phosphors. All these excitations allow us to obtain the red component of the Mn^2+^ emission in the optical spectra [4], which is in accordance with the CIE coordinates of the emission. The emission intensity rises with the increase in the Mn^2+^ concentration, reaching a maximum at *x* = 0.7, and does not increase further until *x* = 1.0. The deviation of the emission intensity from the linear dependence on the Mn^2+^ concentration indicates the onset of concentration quenching.

The PL spectra show a slight shift of the barycenter of the peak from 645 nm to 650 nm with an increase in *x*. This shift means growing of the ^4^T_1_ level excited energy with a weakening of the crystal field, and an increment of structural defects number surrounding the MnO_6_ octahedra. 

Furthermore, the growing influence of these defects is indicated by the increase in the anisotropy of the bond lengths of the *M*5 octahedra beyond *x* > 0.5, which we detected by the PXRD method. These defects, according to [32], reduce the effective value of the crystal field, which leads to a redshift of the luminescence band at *x* > 0.5. Corresponding to the equal occupation of the *M*5 site by Zn^2+^ and Mn^2+^, the same value of *x* = 0.5 in the Ca_9_Zn_1–*x*_Mn*_x_*Na(PO_4_)_7_ host is the limiting concentration of the Mn^2+^ ions, below which the concentration quenching of luminescence does not occur. Taking into account that the distance between the *M*5 sites in the β-Ca_3_(PO_4_)_2_-type structure is 10.4–10.5 Å, we obtain the distance between the luminescent Mn^2+^ ions to be about 21 Å for *x* = 0.5, the last point before concentration quenching. This certainly satisfies the condition of the absence of concentration quenching due to the direct interaction between ions (5–7 Å), but cannot eliminate it due to quenching of structural defects. 

## 5. Conclusions

In summary, the combination of structural and optical characteristics of the Ca_9_Zn_1**−***x*_Mn*_x_*Na(PO_4_)_7_ samples obtained in a “soft” reducing atmosphere using activated carbon indicates the existence of a single-phase solid solution in the 0 ≤ *x* ≤ 1 range. The crystal lattice of the Ca_9_Zn_1**–***x*_Mn*_x_*Na(PO_4_)_7_ solid solution slightly expands with the increase in *x*, when the larger Mn^2+^ ion is substituted with the smaller one (Zn^2+^). According to Rietveld refinement, the zinc and manganese cations, jointly occupying octahedral positions, form the shortest bonds with the surrounding oxygen atoms. Upon excitation at 406 nm, the PL spectra consist of a broad band in the red region due to the ^4^T_1_ → ^6^A_1_ electron transition of Mn^2+^. Defect-related bands at 500–600 nm in the PL spectra are observed only for samples with a low Mn^2+^ content (*x* ≤ 0.2) in the Ca_9_Zn_1**–***x*_Mn*_x_*Na(PO_4_)_7_ compounds. The intensity of the luminescence linearly increases with *x* until *x* > 0.5, when the concentration quenching is revealed to be retarding the growth of the band of luminescence at 650 nm. The onset of this negative effect corresponds to the disordering of the β-Ca_3_(PO_4_)_2_-type structure at *x* > 0.5, which can be seen from the X-ray diffraction analysis. On the contrary, below *x* = 0.5, the large distance between Mn^2+^ in the different *M*5 sites is the reason for the absence of the concentration quenching of luminescence. These results provide important insights into the optical and structural properties of the Ca_9_Zn_1–*x*_Mn*_x_*Na(PO_4_)_7_ solid solution.

## Figures and Tables

**Figure 1 materials-16-04392-f001:**
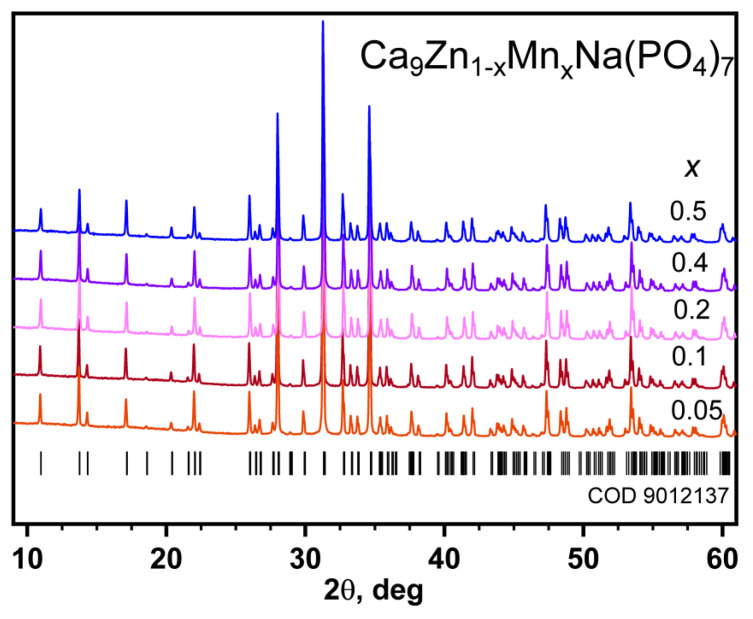
Parts of PXRD patterns of Ca_9_Zn_1**−***x*_Mn*_x_*Na(PO_4_)_7_. Bragg reflections for Ca_9.5_Mg(PO_4_)_7_ (COD 9012137) are shown.

**Figure 2 materials-16-04392-f002:**
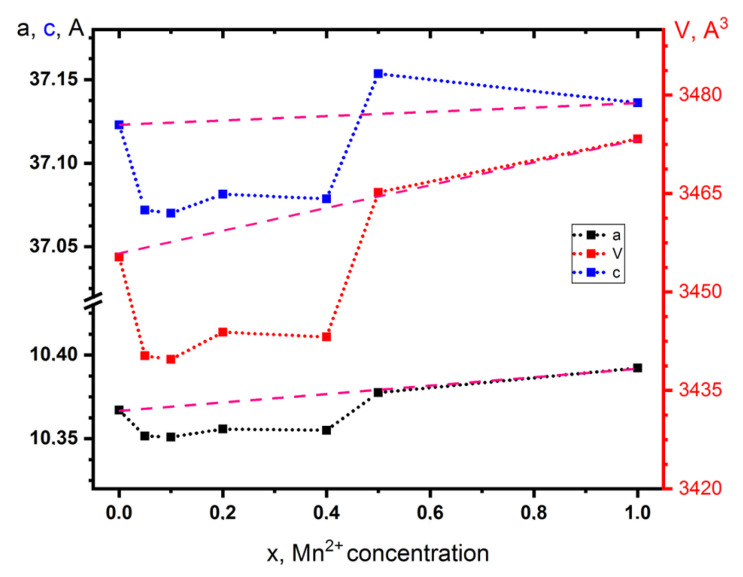
Unit cell parameters *a*, *c* and volume *V* as a function of *x* in Ca_9_Zn_1**–***x*_Mn*_x_*Na(PO_4_)_7_ samples. Pink dash line complies with the Vegard’s law.

**Figure 3 materials-16-04392-f003:**
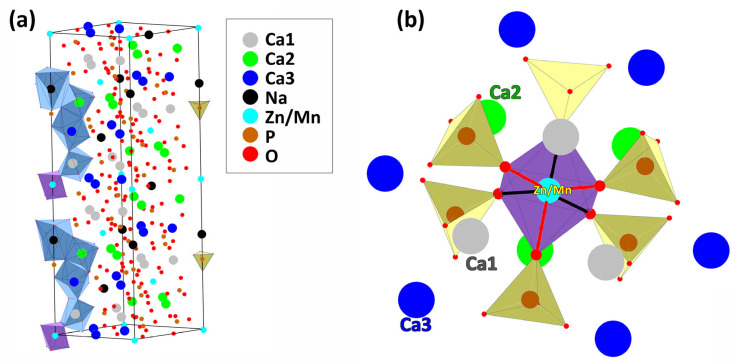
The stereoscopic view of Ca_9_Zn_0.5_Mn_0.5_Na(PO_4_)_7_ structure (**a**). The view of the (Zn/Mn)O_6_ octahedra surrounding along the *c*-axis (**b**).

**Figure 4 materials-16-04392-f004:**
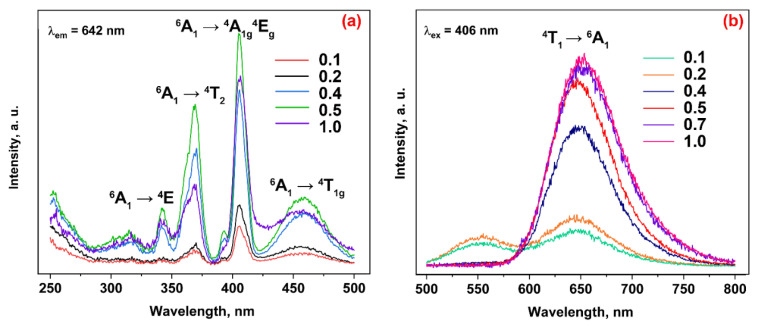
PLE (**a**) and PL (**b**) spectra of Ca_9_Zn_1–*x*_Mn*_x_*Na(PO_4_)_7_.

**Figure 5 materials-16-04392-f005:**
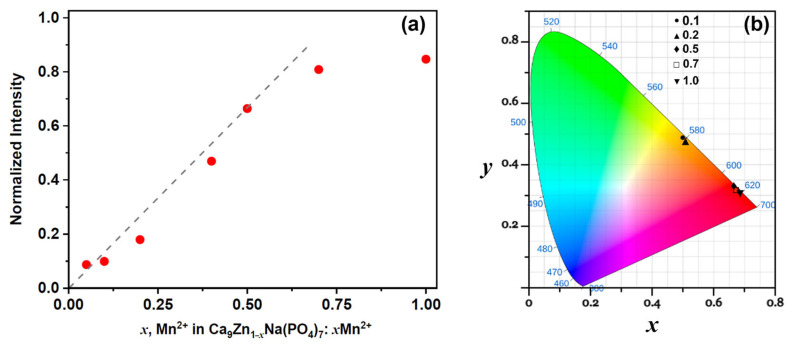
The dependence of PL intensity (**a**) and calculated CIE coordinates (**b**) of Ca_9_Zn_1–*x*_Mn*_x_*Na(PO_4_)_7_ samples (λ_ex_ = 406 nm).

**Table 1 materials-16-04392-t001:** Mean interatomic distances in Ca_9_Zn_1**−***x*_Mn*_x_*Na(PO_4_)_7_ polyhedra.

Mean Distance (Å)	0 *	0.05	0.10	0.20	0.40	0.50	1.0 **
<*M*1-O>	2.50	2.52	2.50	2.50	2.50	2.51	2.48
<*M*2-O>	2.56	2.47	2.47	2.47	2.46	2.47	2.47
<*M*3-O>	2.65	2.57	2.59	2.58	2.57	2.59	2.56
<*M*4-O>	2.50	2.54	2.51	2.56	2.58	2.57	2.70
<*M*5-O>	2.23	2.19	2.22	2.18	2.16	2.25	2.17
<P1-O>	1.65	1.55	1.56	1.55	1.55	1.56	1.56
<P2-O>	1.58	1.53	1.54	1.53	1.54	1.53	1.61
<P3-O>	1.61	1.54	1.54	1.57	1.55	1.53	1.62

* Ca_9.5_Zn(PO_4_)_7_ (PDF#00-048-1148) [25]. ** Ca_9_MnNa(PO_4_)_7_ [11].

**Table 2 materials-16-04392-t002:** *M*–O_6_ and *M*–O_9_ bond lengths around *M*5 position and their differences ∆*l* in Ca_9_Zn_1**−***x*_Mn*_x_*Na(PO_4_)_7_.

*x*	0 *	0.05	0.1	0.2	0.4	0.5	1.0 **
*l* (Zn, Mn–O_6_), Å	2.292	2.155	2.201	2.183	2.154	2.226	1.975
*l* (Zn, Mn–O_9_), Å	2.162	2.217	2.229	2.186	2.166	2.268	2.368
∆*l*, Å	0.13	0.062	0.028	0.003	0.012	0.042	0.393

* Ca_9.5_Zn(PO_4_)_7_ (PDF#00-048-1148) [25]. ** Ca_9_MnNa(PO_4_)_7_ [11].

## Data Availability

Data are available from the corresponding author on reasonable request.

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
