# Peer review of "Mn2+ Luminescence in Ca9Zn1–xMnxNa(PO4)7 Solid Solution, 0 ≤ x ≤ 1"

_materials, 2023, doi:10.3390/ma16124392_

Round 1

Reviewer 1 Report

The present study introduces a very interesting investigation regarding the stabilization of the Mn(II) activator in a calcium phosphate-type matrix in order to produce emission with high relative intensity in the red region, without the presence of co-dopants. Within the area of application of luminescent materials, improving systems that involve environmentally friendly matrices actually arouses interest in the community today. However, for this manuscript to become publishable, some points must be seriously considered and reviewed by the authors, which are explained below.

1. Introduction 

Page 2:

lines 46-49: In order to make the text more user-friendly for a reader unfamiliar with the structure in question, as well as its specific nomenclature, it would be appropriate to define the M1-M5 sites here. And to illustrate by mentioning that the representation of this structure is presented ahead, in Figure 3.  A suggested text is given below.

"In the  the crystal structure of the whitlockite β-Ca3(PO4)2 (TCP), there are five crystallographic independent cation sites (M) inTCP, each of the M(1), M(2), M(3), and M(5) sites are fully occupied by one Ca atom." (J. Phys. Chem. C 2015, 119, 29, 16853–16859 https://doi.org/10.1021/acs.jpcc.5b04997)

line 65: REE, Rare Earth Element (First time being quoted, the acronym has to be elucidated.)

line 84-88: Here authors introduce one of the most important reasons for the relevance of the study. It would be interesting if the authors included in the abstract a mention of this strategic choice of Zn(II) in the composition. It would bring an additional differential to the investigated system. 

2. Materials and Methods

2.3. Photoluminescence study

Regarding the luminescence measurements, it would be important for the authors to inform the type of geometry used in the measurements, that is, if the powders of the samples had their spectra recorded in front face mode or not. And if filter was used to avoid the generation of harmonic signal in the excitation spectra.

3. Results 

3.1. Structural study

Change the sentence "It might be also supposed that Mn2+ and Zn2+ the share common M5 sites in the b-Ca3(PO4)2-type structure." to: It can also be assumed that Mn2+ and Zn2+ share common M5 sites in the b-Ca3(PO4)2-type structure"

page 4

line 139: The phenomenon of X-ray diffraction does not generate "spectra". Here you actually have diffractograms, the powder method, which record diffraction patterns. Please correct this conceptual error.

line 139-140: It is not trivial to locate the structure cards. Authors must provide in the form of supplementary material, so that readers can easily access the refinement profiles and check the figures of merit related to the refinement performed. This is the only way to evaluate the quality of the data obtained and to have reliability in the discussions generated. 

Figure 1 caption needs to be revised.

Line 145: Probably the authors here meant "Vegard's law.

Tables 1 and 2: Distance unit is missing. In Table 2 there is a letter "A" in the first column, however if it is Angstrom, the symbol is incorrect.

3.2. Photoluminescence study

Page 7

line 208: "the series of band" change to: a series of bands are observed

line 213: What is the argument to justify the absence of the CTB? Is it in fact absent or displaced, to the point of not fitting into the measurement range? The authors have to try to give an explanation for this fact.

Page 8

line 225: Besides the fact that saying lines are sharp is redundant, not observing them does not exclude the possibility that there are species of this nature. It's just that they don't reach the level of detection. For exclusion, measurements must be provided, for example, of XPS of these samples, If this is not possible, the authors must soften the statements, they must be conditioned to the level of detection. It would also be interesting to cite references where Mn(IV) emission is observed under similar conditions, to further support the arguments.

4. Discussion

  The placement of the last item as "discussion" is unusual. This content should be distributed throughout the results, deepening the discussion as soon as the data is available. And finally, a "conclusion" item should end the article, summarizing the main points that the study brought.

line 245: Authors state that: "Based on this, one may conclude that all manganese ions in Ca9Zn1-xMnxNa(PO4)7 were completely incorporated into the b-Ca3(PO4)2-type structure"The χ2 and  Rwp parameters from Rietiveld Refinement could bring more reliability in this statement.

Finally, relative or absolute quantum yield data would be very relevant so that the authors could establish the state of the art of the proposed system compared to those reported in the literature.

The observed corrections were already mentioned in the previous item.

Reviewer 2 Report

The stable temperature of stoichiometric mixture should be mentioned.

Can we prove incorporation of Mn+2 in Ca9Zn1- 245 xMnxNa(PO4)7 by any other technique? other than PXRD.  

Quality of English is ok however, few grammatical mistakes need to be addressed.

Reviewer 3 Report

In this work, Mn2+ luminescence with no concentration quenching in Ca9Zn1-xMnxNa(PO4)7 solid solution, 0.05 < x < 0.5. The idea of this research is interesting to readers. The background is well studied and the presentation of the method is very clear and sound, but there are some minor issues to be addressed:

Comments:

Title should be revised with meaningful.

The author should provide the more quantitative information in the abstract section.

The following section should be provided the suitable reference. 2.1. Synthetic Procedure.

The red luminescence color evidence should be provided.

Line 207; Figure 4 shows PLE (a) and PL (b) spectra of Ca9Zn1–xMnxNa(PO4)7 compounds. On the PLE spectra monitored at λex = 642 nm. The author should confirm it. λex or λem?

There is no references in the discussion section. The author should provide the suitable references in discussion section.

Conclusion section should be provide separately with outstanding point of this work.

Typographical errors and superfluous spaces throughout the manuscript should be corrected.

Round 2

Reviewer 1 Report

The authors promoted the suggested changes and satisfactorily argued the questions. In this way, it is recommended to accept the article as it is.